

# A new derivation of the relationship between diffusion coefficient and entropy in classical Brownian motion by the ensemble method

Yi Liao[1*] and Xiao-Bo Gong[2,3,4†]

**1** Department of Physics, College of Science, Southern University of Science and Technology, Shenzhen, 518055, China
**2** Yunnan Observatory and Key Laboratory for the Structure and Evolution of Celestial Objects, Chinese Academy of Sciences, Kunming, 650011, China
**3** Center for Astronomical Mega-Science, Chinese Academy of Sciences, 20A Datun Road, Chaoyang District, Beijing, 100012, China
**4** University of Chinese Academy of Sciences, Beijing, 100049, China

★ liaoyitianyi@gmail.com, † gxbo@ynao.ac.cn

## Abstract

The diffusion coefficient–a measure of dissipation, and the entropy–a measure of fluctuation are found to be intimately correlated in many physical systems. Unlike the fluctuation dissipation theorem in linear response theory, the correlation is often strongly non-linear. To understand this complex dependence, we consider the classical Brownian diffusion in this work. Under certain rational assumption, i.e. in the bi-component fluid mixture, the mass of the Brownian particle $M$ is far greater than that of the bath molecule $m$, we can adopt the weakly couple limit. Only considering the first-order approximation of the mass ratio $m/M$, we obtain a linear motion equation in the reference frame of the observer as a Brownian particle. Based on this equivalent equation, we get the Hamiltonian at equilibrium. Finally, using canonical ensemble method, we define a new entropy that is similar to the Kolmogorov-Sinai entropy. Further, we present an analytic expression of the relationship between the diffusion coefficient $D$ and the entropy $S$ in the thermal equilibrium, that is to say, $D = \frac{\hbar}{eM} \exp[S/(k_B d)]$, where $d$ is the dimension of the space, $k_B$ the Boltzmann constant, $\hbar$ the reduced Planck constant and $e$ the Euler number. This kind of scaling relation has been well-known and well-tested since the similar one for single component is firstly derived by Rosenfeld with the expansion of volume ratio.

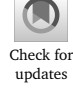
# 1 Introduction

Study of relationship between diffusion coefficient $D$ of a tagged molecule and the entropy $S$ of complex systems has been an interesting topic in statistical physics since the first quantitative relation between the two was established by Adam and Gibbs [1]. It provides a good viewpoint to access the field of the Brownian motion in some complex fluid [2].

In 1977, the scaling relationship between diffusion coefficient and the excess entropy of single component, which only includes the Brownian particle, which reads $D = a \exp(bS/k_B)$, where $a$ and $b$ only are some empirical fitting parameters and $k_B$ is the Boltzmann constant, was first proposed by Rosenfeld with the expansion of volume ratio [3, 4]. The scaling relationship reads

$$D^* = D \frac{\rho^{1/3}}{(k_B T/m)^{1/2}} \equiv a \exp(bS_{ex}/k_B), \tag{1}$$

where $S_{ex} = \frac{S_{tot} - S_I}{N}$, $S_{tot}$ is the total entropy of the system, $S_I$ is the entropy of the ideal gas, $a$ and $b$ are the empirical fitting parameters, $m$ is atom mass, $\rho$ is the number density. And Dzugutov proposed a similar universal scaling relationship, where the entropy is defined through the radial distribution function [5]. These relationships have been well-tested by many experiments in different systems [6–10]. The scaling relationship reads

$$D^* = \frac{D}{4\sigma^4 g(\sigma)\rho(\pi k_B T/m)^{1/2}} \equiv a \exp(bS_{ex}/k_B), \tag{2}$$

where $\sigma$ is the hard-sphere diameter, $g(\zeta)$ is the radial distribution function. In real system, $\sigma$ is the position of the maximum of the function $g(\zeta)$.

A more rigorous scaling law for the binary fluid mixture has been presented at the beginning of 21th century [11, 12]. However, in Ref. [11], the entropy is defined in thermodynamic form and dependent on the partition function. The kind of canonical entropy is hard to analytically calculate. And one has to make the cut-off in the cluster expansion to calculate it. In Ref. [11], the result of the entropy is only at the level of the two-body interaction accuracy. All above-mentioned universal scaling laws are found to fail in low density case due to the parameter $b$ varying [13]. In the binary fluid mixture, the mass dependence of diffusivity happens [14, 15]. Considering that the mass of Brownian particles, such as colloids, is far heavier than one of bath particles, we aim at this kind of relationship in low density case in this paper. Using the canonical ensemble method, we define a new entropy that is similar to the Kolmogorov-Sinai entropy. The definition of Kolmogorov-Sinai entropy is based on the change ratio of phase-space volume as time varying so that it is easier to calculate than the thermodynamic entropy. At the accuracy of the first-order approximation of the mass ratio, we present a analytic expression of the relationship between the diffusion coefficient $D$ and the

entropy $S$ in the thermal equilibrium where the parameter $a$ and $b$ are explicitly given. Hereunto, although the Rosenfeld's relationship seemly does not have an acknowledged theoretical explanation [16], we try to provide an alternative view to interpret it in this work.

The outline of this paper is as follows. In Section II, we consider the classical Brownian diffusion. Under certain rational assumption, i.e. in the bi-component fluid mixture, the mass of the Brownian particle $M$ is far greater than that of the bath molecule $m$, we know that every Browian particle suffer the same stochastic force. In Section III, we obtain a linear motion equation in the reference frame of the observer as a Brownian particle and give the Hamiltonian at equilibrium. In Section IV, using these snapshot probability distributions, we define a new entropy and present the relationship between diffusion coefficient and entropy. Finally, in Section V, to check the superiority of our treatment, we compare our results with that of hard-sphere model where the entropy is dependent of the volume ratio.

## 2 Langevin equation and Langevin operator

A Brownian motion particle in $d$-dimensional space can be described by the Langevin equation

$$M\frac{d^2\mathbf{x}}{dt^2} + \alpha\frac{d\mathbf{x}}{dt} = \zeta(t), \tag{3}$$

where $M$ is the particle mass, $\zeta(t)$ is the white Gaussian noise with correlations $\langle\zeta_i(t)\zeta_j(t')\rangle = 2\alpha k_B T\delta_{ij}\delta(t-t')$. The diffusion coefficient $D$ satisfies Einstein's relation $D = \frac{k_B T}{\alpha}$. The velocity has a decay time $\gamma^{-1}$, where $\alpha = M\gamma$. In general, the mass of the particle is very small in micro/nano scale. The inertial term can be ignored, compared with the viscosity term. That is in the low Reynolds number regime where the Stoke-Einstein relation could be established. When the system is at equilibrium, the total entropy production rate is zero, and the velocity of Brownian particles follows the Maxwell-Boltzmann velocity distribution [17, 18]. There exits many techniques to obtain the Langevin equation [19, 20]. One of these techniques is as follows [19]. Considering a system including $N$ light bath molecules of mass $m$ and a heavy point-like Brownian particle of mass $M$, the mass ratio $\lambda^2 = \frac{m}{M}$ is very small. The Hamiltonian of the system is

$$H_s = \frac{1}{2M}\mathbf{p}^2 + H_0, \tag{4}$$

$$H_0 = \frac{\mathbf{p}^N \cdot \mathbf{p}^N}{2m} + U(\mathbf{r}^N) + \Phi(\mathbf{r}^N, \mathbf{x}), \tag{5}$$

where $\mathbf{p}$ is the momentum of the Brownian particle, $\mathbf{p}^N$ and $\mathbf{r}^N$ are $Nd$-dimensional positions and momentums of the bath molecules, $U(\mathbf{r}^N)$ is the two-body interaction potential between bath molecules, $\Phi$ is the interaction potential between bath molecules and the Brownian particle. The Liouville operator $L$ is defined by

$$\begin{aligned} L &= L_0 + L_1, \\ L_0 &= \frac{\mathbf{p}^N}{m}\cdot\nabla_{\mathbf{r}^N} - \nabla_{\mathbf{r}^N}H_0\cdot\nabla_{\mathbf{p}^N}, \\ L_1 &= \frac{\mathbf{p}}{M}\cdot\nabla_{\mathbf{x}} - \nabla_{\mathbf{x}}\Phi\cdot\nabla_{\mathbf{p}} = \lambda(\frac{\overline{\mathbf{p}}}{m}\cdot\nabla_{\mathbf{x}} - \nabla_{\mathbf{x}}\Phi\cdot\nabla_{\overline{\mathbf{p}}}) = \lambda L_2, \end{aligned} \tag{6}$$

where $\overline{\mathbf{p}} = \lambda\mathbf{p}$. The projection operator $\hat{P}$ is defined by the following equation [19]

$$\hat{P}A = \langle A\rangle = \int Z_0^{-1} e^{-\beta H_0(t=0)} A d\mathbf{r}^N d\mathbf{p}^N, \tag{7}$$

here $\beta \equiv k_B T$, and the partition function $Z_0 = \int e^{-\beta H_0(t=0)} d\mathbf{r}^N d\mathbf{p}^N$. $\zeta(0)$ indicates the force at $t = 0$. Then we can get $\zeta(t) = e^{Lt}\zeta(0), \langle \zeta(0) \rangle = 0$. Finally, as was shown in Refs. [18, 19, 21], the Langevin equation is given by

$$
\begin{aligned}
\frac{d\overline{\mathbf{p}}}{dt} &= \lambda^2 \int_0^t e^{L(t-\tau)} \hat{P} L_2 \zeta^+(\tau) d\tau + \lambda \zeta^+(t) \\
&= \lambda^2 \int_0^t e^{L(t-\tau)} (\nabla_{\overline{\mathbf{p}}} - \beta \frac{\overline{\mathbf{p}}}{m}) \cdot \langle \zeta(0) \zeta^+(\tau) \rangle d\tau + \lambda \zeta^+(t) \\
&\approx -\lambda^2 \frac{\beta}{m} \int_0^t \overline{\mathbf{p}}(t-\tau)) \cdot \langle \zeta(0) \zeta_0(\tau) \rangle d\tau + \lambda \zeta_0(t) \\
&= -\gamma \overline{\mathbf{p}} + \lambda \zeta_0(t),
\end{aligned}
\tag{8}
$$

here $\zeta^+(t) = e^{\hat{O}Lt}\zeta(0)$ with the operator $\hat{O} = 1 - \hat{P}$, and $\zeta_0(t) = e^{L_0 t}\zeta(0)$. The above equation is obtained in the weak coupling limit( namely, $\lambda^2 \to 0, t \to \infty, \lambda^2 t$ is limited) [19].

## 3 Hamiltonian at equilibrium in the reference frame of the observer as a Brownian particle

Because the mass of the Brownian particle is far greater than that of the bath molecules (i.e. $M \gg m$), the mean velocity of Brownian particle is far slower than that of the bath molecules, the force on an arbitrary Brownian particle approximately equals to $\zeta_0(t)$ [18]. One can choose a Brownian particle as an observer which has the same initial position as the Brownian particles is motionless at $t = 0$. The sign $\nu_0$ indicates the initial velocity of a Brownian particle. The position $x^o$ of the observer satisfies the Langevin equation

$$
M\frac{d^2\mathbf{x}^o}{dt^2} + \alpha \frac{d\mathbf{x}^o}{dt} = \zeta(t).
\tag{9}
$$

For convenience, we introduce $\mathbf{y} \equiv \mathbf{x} - \mathbf{x}^o$. In the reference frame of the observer, $\mathbf{y}$ satisfies the equation which reads,

$$
M\frac{d^2\mathbf{y}}{dt^2} + M\gamma \frac{d\mathbf{y}}{dt} = 0,
\tag{10}
$$

where, the initial relative position is zero and the initial relative velocity $\nu_0$. Its solution is $\mathbf{y} = \frac{\nu_0}{\gamma}(1 - e^{-\gamma t})$. The solution also satisfies an other system that is described by [22]

$$
M\frac{d^2\mathbf{y}}{dt^2} \equiv -\frac{\partial \phi(\mathbf{y})}{\partial \mathbf{y}} = M\gamma^2 \mathbf{y} - M\gamma \nu_0,
\tag{11}
$$

here the potential reads $\phi(\mathbf{y}) = \text{constant} + M\gamma \nu_0 \cdot \mathbf{y} - \frac{M}{2}\gamma^2 \mathbf{y}^2$. Both systems share the common phase curve, thus we can get

$$
\phi(\mathbf{x} - \mathbf{x}^o) \approx \phi(\mathbf{0}) + M\gamma \nu_0(\mathbf{x} - \mathbf{x}^o) - \frac{M}{2}\gamma^2(\mathbf{x} - \mathbf{x}^o)^2.
\tag{12}
$$

Eq.(8) is a second-order equation of $\lambda$ and $\phi$ is the same level. Now the system is linear and will reach equilibrium at $t = \infty$. Two particles with the same initial position but a initial velocity difference $\nu_0$ can get a maximum divergence of $\Delta \mathbf{x} = \frac{\nu_0}{\gamma}$, therefore the term $M\gamma \nu_0(\mathbf{x} - \mathbf{x}^o)$ will involve in the form being $M(\gamma \Delta \mathbf{x})^2$. Consequently, the final Hamiltonian of the ensemble system with $n$ Brownian particles in the reference frame at equilibrium reads

$$
H_{\text{total}}(t = \infty) = \sum_i^n \left[ \frac{1}{2M}\mathbf{p}_i^2 + \frac{M}{2}\gamma^2(\mathbf{x}_i - \mathbf{x}^o)^2 + \phi(\mathbf{0}) \right].
\tag{13}
$$

It needs to point that the entropy whose definition depends upon the Hamiltonian is similar to the Kolmogorov-Sinai entropy. The definition of Kolmogorov-Sinai entropy is based on the change ratio of phase-space volume as time varying. Dzugutov, Aurell and Vulpiani have made the assumption that the Kolmogorov-Sinai entropy can be connected to the conventional thermodynamic entropy [23]. The derivation of Eq.(13) based on the Kolmogorov-Sinai entropy would be showed in APPENDIX A. In APPENDIX B, the formula of the thermodynamic entropy of Brownian particle is derived, but it is hard to analytically solve. Fortunately, Dzugutov et al. have point that Kolmogorov-Sinai entropy, when expressed in terms of the atomic collision frequency, is uniquely related to the thermodynamic excess entropy by a universal linear scaling law [23]. The linear law is not influence the exponential relationship between the diffusion coefficient and the entropy.

## 4   Relationship between diffusion coefficient and entropy

When the system is in the thermal equilibrium, we can use the canonical ensemble method to calculate the entropy. Form Eq.(13), one can know that the system is uncouple. The one-particle partition function

$$Z = \frac{1}{(2\pi\hbar)^d} \int \exp\left\{-\beta\left[\frac{1}{2M}\mathbf{p}^2 + \frac{M}{2}\gamma^2(\mathbf{x}-\mathbf{x}^o)^2 + \phi(0)\right]\right\} d\mathbf{p}\,d\mathbf{x} = \left(\frac{1}{\hbar\beta\gamma}\right)^d e^{-\beta\phi(0)}, \quad (14)$$

here $\hbar$ is the reduced Planck constant. The one-particle entropy is

$$S = k_B\left(\ln Z - \beta\frac{\partial}{\partial\beta}\ln Z\right) = k_B d[1 - \ln(\hbar\beta\gamma)] = k_B d\ln\left(\frac{eMD}{\hbar}\right), \quad (15)$$

here, $e$ is the Euler number. The relationship between diffusion coefficient and entropy reads,

$$D = \frac{\hbar}{eM}\exp[S/(k_B d)] \equiv a\exp(bS/k_B), \quad (16)$$

here the parameter $a = \frac{\hbar}{eM}$ and $b = \frac{1}{d}$.   In an anisotropic system, if the particle has the corresponding diffusion coefficient $D_i$ in the different dimension, one can get

$$\prod_{i=1}^d D_i = (\frac{\hbar}{eM})^d \exp(S/k_B). \quad (17)$$

## 5   Results and discussion

Our result shown in Eq.(16) has the same form as Eq.(1)and Eq.(2), but our method can give the analytic formula and make it possible to calculate some more complex model.

In this paper, we only consider the point-like particle and the accuracy of the $\lambda^2$. To obtain the more accurate relationship, one can expand the motion equation in the higher-order terms of $\lambda$. The entropy can be expanded in terms of $\lambda$ related to the mass ratio. $\lambda$ maybe plays the same role as the quantity related to the volume ratio,such as $\eta$ in the 3-dimensional hard-sphere model.   In the model that has been well-solved at the level of 10-body interaction, the entropy is [3]

$$S = Nk_B\left[\ln\left(\frac{2\pi mk_BT}{h^2}\right)^{\frac{3}{2}} + \frac{5}{2} + \ln\frac{1}{\rho} - \frac{4\eta - 3\eta^2}{(1-\eta)^2}\right], \quad (18)$$

where $\eta = \frac{\pi N \overline{d}^3}{6V}$, $\overline{d}$ is the hard-sphere diameter, $S_{ex} = -\frac{4\eta - 3\eta^2}{(1-\eta)^2}$. Because diffusion coefficient $D \propto \frac{\overline{v}}{\rho \overline{d}^2}$, and $(\rho)^{-\frac{1}{3}}$ is larger than $\overline{d}$ for the dilute gas, so that $b$ is larger than $\frac{1}{3}$ for the function $D = a \cdot e^{b \cdot s / k_B}$. For Brownian particle, its mass and volume is far lager than that of bath molecules, its remaining space is filled with these light molecules, so that its $(\rho)^{-\frac{1}{3}}$ is close to $\overline{d}$, then Eq.(16) is roughly right. On the other hand, Eq.(8) is only valid up to order $\lambda^2$. the term $S_{ex}$ will be included in the nonlinear Langevin equation

$$M\frac{d^2 x}{dt^2} + \alpha \frac{dx}{dt} + \alpha_1 \left(\frac{dx}{dt}\right)^3 = \zeta(t), \tag{19}$$

where $\frac{\alpha_1}{\alpha} \approx \frac{m}{6k_B T}$ for the generalized Rayleigh model [18]. The relationship in the nonlinear Langevin equation will be considered in our future work.

## Acknowledgments

Y. Liao would thank Prof. Miao Li. This work was supported in part by his startup funding of the Southern University of Science and Technology. X.-B. Gong would thank Prof. Feng-Hui Zhang. This work was also supported by National Natural Science Foundation of China (NSFC) under grants (Y6GJ161001).

## A Definition of Kolmogorov-Sinai entropy and derivation of final Hamiltonian

We introduce the Kolmogorov-Sinai entropy defined as

$$S = \sup_{Q} h(Q) \equiv \sup_{Q} \left\{ -\lim_{n \to \infty} \frac{1}{n\tau} \sum_{\omega} \mu(\omega) \ln \mu(\omega) \right\},$$
$$\omega = \{\mathbf{X}(t) = (t_i, \mathbf{X}_i), t_i = i\tau, i = 0, 1, 2 \cdots, n-1\}. \tag{1}$$

Here, $\omega$ denotes a path of the particle, and $\mu(\omega)$ is the probability. A Brownian motion particle can be described by the Langevin equation which reads

$$\frac{d\mathbf{p}}{dt} + \gamma \mathbf{p} = \zeta_0(t), \zeta_0(t) = e^{L_0 t} \zeta(0). \tag{2}$$

In the 1st and 2nd ensemble, the path of the Brownian particle is $\mathbf{X}_{00}(t)$ and $\mathbf{X}_{11}(t)$, respectively. In the two ensembles, the Brownian particles have different initial velocities being $\mathbf{v}_{00}$ and $\mathbf{v}_{11}$, but the bath molecules have the same initial velocity distributions. Due to $\zeta_0(t) = e^{L_0 t} \zeta(0)$, the forces $\zeta_0(t)$ are the same. One can get

$$\lim_{n \to \infty} [\mathbf{X}_{11}(t) - \mathbf{X}_{00}(t)] = \lim_{t \to \infty} [\mathbf{X}_{11}(t) - \mathbf{X}_{00}(t)] = \frac{\mathbf{v}_{11} - \mathbf{v}_{00}}{\gamma}. \tag{3}$$

Assuming that the systems are in thermal equilibrium, one can get the probability ratio of two paths which reads

$$\frac{\mu(\omega_2)}{\mu(\omega_1)} \propto \exp\left[-\frac{M(\mathbf{v}_{11} - \mathbf{v}_{00})^2}{2k_B T}\right] = \lim_{t \to \infty} \exp\left\{-\frac{M\gamma^2 [\mathbf{X}_{11}(t) - \mathbf{X}_{00}(t)]^2}{2k_B T}\right\}. \tag{4}$$

So, when $t \to \infty$, the probability of all possible paths satisfies

$$\mu \propto \exp\left\{ -\frac{M\mathbf{v}_{00}^2}{2k_B T} - \frac{M\gamma^2[\mathbf{X}_{11}(t) - \mathbf{X}_{00}(t)]^2}{2k_B T} \right\}. \tag{5}$$

Therefore, based on the definition of Kolmogorov-Sinai entropy, one can obtain the final Hamiltonian of the ensemble system which reads

$$H_{\text{total}}(t = \infty) = \sum_{i}^{n}\left[ \frac{1}{2M}\mathbf{p}_i^2 + \frac{M}{2}\gamma^2(\mathbf{x}_i - \mathbf{x}^o)^2 + \phi(0) \right]. \tag{6}$$

## B Formula of the thermodynamic entropy of Brownian particle

One can assume that a system labelled as System 1 with the volume $V$, only includes $N$ bath particles, which Hamiltonian reads,

$$H = \frac{\mathbf{p}^N \cdot \mathbf{p}^N}{2m} + U(\mathbf{r}^N). \tag{7}$$

The partition function of this system under canonical ensemble is

$$Z_1 = \frac{1}{N!h^{dN}} \int e^{-\beta H} d\mathbf{p}_1...d\mathbf{p}_N d\mathbf{r}_1...d\mathbf{r}_N. \tag{8}$$

When one introduces a heavier Brownian particle to join in the system, it is labelled as System 2, which partition function is

$$Z_2 = \frac{1}{N!h^{dN}h^d} \int e^{-\beta H_s} d\mathbf{p}_1...d\mathbf{p}_N d\mathbf{r}_1...d\mathbf{r}_N d\mathbf{p}d\mathbf{x}, \tag{9}$$

one can define the entropy of Brownian particle which equals the difference of entropy of System 2 and System 1. One can obtain $\Delta \ln Z$, which reads

$$\begin{aligned}
\Delta \ln Z &\equiv \ln Z_2 - \ln Z_1 \\
&= \frac{d}{2}\ln\left(\frac{2\pi M}{h^2\beta}\right) + \ln\left(\int e^{-\beta\Phi - \beta U} d\mathbf{r}_1...d\mathbf{r}_N d\mathbf{x}\right) - \ln\left(\int e^{-\beta U} d\mathbf{r}_1...d\mathbf{r}_N\right) \\
&= \frac{d}{2}\ln\left(\frac{2\pi M}{h^2\beta}\right) - \ln\left(\frac{\langle e^{\beta\Phi}\rangle}{V}\right).
\end{aligned} \tag{10}$$

Based on the formula of the thermodynamic entropy being

$$S = k\left(\ln Z - \beta\frac{\partial}{\partial\beta}\ln Z\right). \tag{11}$$

The thermodynamic entropy of Brownian particle $S_T$ reads

$$S_T = \frac{kd}{2}\left[\ln\left(\frac{2\pi M}{h^2\beta}\right) + 1\right] - k\ln\left[\frac{\langle e^{\beta\Phi}\rangle}{V}\right] + k\beta\frac{\partial}{\partial\beta}\ln(\langle e^{\beta\Phi}\rangle). \tag{12}$$

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
