# Peer review of "A new derivation of the relationship between diffusion coefficient and entropy in classical Brownian motion by the ensemble method"

_SciPost Physics, doi:SciPost Phys. Core 4, 015 (2021)_

## Round 1 · Referee Report · Anonymous (Referee 1) · 2020-12-8

Report

The 1977 work of Rosenfeld exerted a strong influence on the literature of computer simulation of liquids, which is however accessible only the researchers with expertise in this specific field.
The issue of entropy and transport is of more general interest.

More recently, in the late 90’s there has been some attention devoted to explore the role of Komogorov-Sinai entropy in the dynamics of liquid. This corresponds to a transition from the mere technical level to a more important level addressing the fundamental principle of statistical mechanics.

The authors focus on the microscopic derivation of the Langevin equation with the method of Mazur and Oppenheim, which is a sort projection method. They derive the Langevin equation and address the important issue of establishing the correct Hamiltonian structure corresponding to this Langevin equation. To do that they define a reference system evaluated by neglecting the stochastic force and establishes the time distance between two Brownian particles that has the same initial position but different velocities. This is central point of their work that is based on the work of Ref. [22]. They define another dynamical system leading to the same solution and sharing a common phase curve. It is not clear to me what is the physical meaning of this statement. On the basis of this crucial technique they define the Hamiltonian of Eq. (11) and adopting the canonical partition function they establish a connection with thermodynamics, the resulting entropy being interpreted by them as a sort of Kolmgorov-Sinai entropy. This is not unreasonable since Dzugutov and Vulpiani, not quoted in the paper, made the assumption that the Kolmogorov-Sinai entropy can be connected to the conventional thermodynamic entropy.

I would be very happy to recommend this paper for publication, if the authors will be able to explain their derivation of Hamiltonian (11).
  • validity: -
  • significance: -
  • originality: -
  • clarity: -
  • formatting: -
  • grammar: -

Author:  Yi Liao  on 2021-01-13  [id 1150]

(in reply to Report 1 on 2020-12-08)
Category:
answer to question
validation or rederivation

Thanks for your kind considerations and referee’s detailed comments and suggestions to improve this work.

We thank the referee's approvement. All the revised parts are written in bold.

In the revised paper, we have explained our derivation of Hamiltonian (11) in APPENDIX A.

We introduce the Kolmogorov-Sinai entropy defined as
\begin{eqnarray}
S&=&\underset{Q}{\sup} h(Q)\equiv \underset{Q}{\sup}\{-\underset{n\rightarrow \infty}{\lim}\frac{1}{n\tau}\underset{\omega}{\sum}\mu(\omega)\ln \mu(\omega) \},
\\\nonumber
\omega&=&\{\textbf{X}(t)=(t_i,\textbf{X}_i),t_i=i\tau,i=0,1,2\cdots,n-1\}.
\end{eqnarray}
Here, $\omega$ denotes a path of the particle, and $\mu(\omega)$ is the probability.
A Brownian motion particle can be described by the Langevin equation which reads
\begin{equation}
\frac{d\textbf{p}}{d t}+\gamma \textbf{p}=\boldsymbol{\zeta}_0(t), \boldsymbol{\zeta}_{0}(t)=e^{L_{0}t}\boldsymbol{\zeta}(0).
\end{equation}
In the 1st and 2nd ensemble, the path of the Brownian particle is $\textbf{X}_{00}(t)$ and $\textbf{X}_{11}(t)$, respectively. In the two ensembles, the Brownian particles have different initial velocities being $\textbf{v}_{00}$ and $\textbf{v}_{11}$, but the bath molecules have the same initial velocity distributions. Due to $\boldsymbol{\zeta}_{0}(t)=e^{L_{0}t}\boldsymbol{\zeta}(0)$, the forces $\boldsymbol{\zeta}_{0}(t)$ are the same. One can get
\begin{equation}
\underset{n\rightarrow \infty}{\lim}[\textbf{X}_{11}(t)-\textbf{X}_{00}(t)]=\underset{t\rightarrow \infty}{\lim}[\textbf{X}_{11}(t)-\textbf{X}_{00}(t)]=\frac{\textbf{v}_{11}-\textbf{v}_{00}}{\gamma}.
\end{equation}
Assuming that the systems are in thermal equilibrium, one can get the probability ratio of two paths which reads
\begin{equation}
\frac{\mu(\omega_2)}{\mu(\omega_1)}\propto \exp[-\frac{M(\textbf{v}_{11}-\textbf{v}_{00})^2}{2k_B T}]=\underset{t\rightarrow \infty}{\lim}\exp\{-\frac{M\gamma^2[\textbf{X}_{11}(t)-\textbf{X}_{00}(t)]^2}{2k_B T}\}.
\end{equation}
So, when $t\rightarrow \infty$, the probability of all possible paths satisfies
\begin{equation}
\mu\propto \exp\{-\frac{M\textbf{v}^2_{00}}{2k_B T}-\frac{M\gamma^2[\textbf{X}_{11}(t)-\textbf{X}_{00}(t)]^2}{2k_B T}\}
\end{equation}
Therefore, based on the definition of Kolmogorov-Sinai entropy, one can obtain the final Hamiltonian of the ensemble system which reads
\begin{equation}
H_{\rm total}(t=\infty)=\underset{i}{\overset{n}{\sum}}[\frac{1}{2M}\textbf{p}_i^{2}+\frac{M}{2}\gamma^{2}(\textbf{x}_i-\textbf{x}^{o})^{2}+\phi(0)].
\end{equation}

Attachment:

A_new_derivation_of_the_relationship_X1ol6ZD.pdf

---

## Round 1 · Referee Report · Anonymous (Referee 2) · 2021-2-16

Report

The authors report a derivation of a relation between
entropy and diffusion coefficient of Brownian point-like
particles using Canonical ensemble.

This is a well studied problem, as the authors have said.
Similar scaling relations are mentioned in Eqs. 16 & 17.
But they have proposed a new way in deriving such scaling,
which could be interesting.

Before recommending, I expect the authors to address the
following points:

Requested changes

  1. What is, if any, the advantage of this method over the other methods? Is there any experimental comparison where this method seem to work better?

  2. The only comparison made so far is in the case of hard sphere model (Eq. 18), where it is argued that for massive Brownian particles (compared to that of the bath molecules), the expressions of entropies are comparable. Is there a quantitative measure of it possible (say, with typical parameter values)?

  3. Finally, the organization of the paper is somewhat confusing. Earlier attempts (Eqs. 16 & 17) should not come in the results section but should be moved to the introduction.

  • validity: high
  • significance: good
  • originality: good
  • clarity: high
  • formatting: reasonable
  • grammar: good

Author:  Yi Liao  on 2021-03-13  [id 1305]

(in reply to Report 2 on 2021-02-16)
Category:
answer to question

Thanks for your kind considerations and referee’s detailed comments and suggestions to improve this work.

We thank the referee's approvement. All the revised parts are written in bold.

Point by point: Reply to the referee comment

The authors report a derivation of a relation between entropy and diffusion coefficient of Brownian point-like particles using Canonical ensemble.

This is a well studied problem, as the authors have said. Similar scaling relations are mentioned in Eqs. 16 & 17. But they have proposed a new way in deriving such scaling, which could be interesting.

Before recommending, I expect the authors to address the following points:

Answer: We thank the referee's approvement. In the revised paper, we have performed more details to address the three points based on the referee's suggestions.

1- What is, if any, the advantage of this method over the other methods? Is there any experimental comparison where this method seem to work better?

Answer: The derivation of Eq.(13) based on the Kolmogorov-Sinai entropy would be showed in APPENDIX A. in APPENDIX B, the formula of the thermodynamic entropy of Brownian particle is derived, but it is hard to analytically solve. Fortunately, Dzugutov et al. have point that Kolmogorov-Sinai entropy, when expressed in terms of the atomic collision frequency, is uniquely related to the thermodynamic excess entropy by a universal linear scaling law (Dzugutov M., Aurell E., and Vulpiani A. 1998, Phys. Rev. Lett. 81, 1762). The linear law is not influence the exponential relationship between the diffusion coefficient and the entropy. Our method can give the analytic formula of Kolmogorov-Sinai entropy and make it possible to calculate some more complex model. The Kolmogorov-Sinai entropy is regarded as a measure, for the loss of information about the state of the system, per unit of time. This quantity is more mathematical than physical, so there are not any experimental comparison.

2- The only comparison made so far is in the case of hard sphere model (Eq. 18), where it is argued that for massive Brownian particles (compared to that of the bath molecules), the expressions of entropies are comparable. Is there a quantitative measure of it possible (say, with typical parameter values)?

Answer: One can assume that a system labelled as System $1$ with the volume $V$, only includes $N$ bath particles,which Hamiltonian reads,

$$ H= \frac{\textbf{p}^{N}\cdot\textbf{p}^{N}}{2m}+U(\textbf{r}^{N}). $$
The partition function of this system under canonical ensemble is
$$ Z_{1}=\frac{1}{N!h^{dN}}\int e^{-\beta H}d\textbf{p}{1}... d\textbf{p}}d\textbf{r{1}...d\textbf{r}.\ $$
When one introduces a heavier Brownian particle to join in the system, it is labelled as System $2$, which partition function is
$$ Z_{2}=\frac{1}{N!h^{dN}h^{d}}\int e^{-\beta H_{s}}d\textbf{p}{1}... d\textbf{p}}d\textbf{r{1}...d\textbf{r} $$}d\textbf{p}d\textbf{x
one can define the entropy of Brownian particle which equals the difference of entropy of System 2 and System 1. One can obtain $\Delta \ln Z\equiv \ln Z_{2}-\ln Z_{1}$, based on the formula of the thermodynamic entropy, the thermodynamic entropy of Brownian particle $S_{T}$ reads
$$ S_{T}=\frac{kd}{2}[\ln(\frac{2\pi M}{h^{2}\beta})+1]- k\ln[\frac{<e^{\beta\Phi}>}{V}]+k\beta \frac{\partial}{\partial\beta }\ln(<e^{\beta\Phi}>). $$
Because
$$ <e^{A}>=<1+A+\frac{1}{2}A^{2}+\frac{1}{6}A^{3}+\ldots> = e^{\langle A\rangle + \frac{1}{2}(\langle A^2 \rangle - \langle A \rangle^2) + O(A^3)} $$

$$ \frac{\partial}{\partial\beta }<\Phi>=<\Phi><H_{0}>-<\Phi H_{0}> =<\Phi>+<\Phi><\Phi>-<\Phi>^{2}-<\Phi U>\ =<\Phi><\Phi>-<\Phi>^{2}\ $$

$$ <\phi(\textbf{x}-\textbf{r}{i}) U(\textbf{r}}-\textbf{r{j})> =<\phi>(first ~integral ~with~\textbf{r}). $$
So, there is a quantitative measure of it possible but still hard if one can know the values of $<\Phi>$ and $$.

3- Finally, the organization of the paper is somewhat confusing. Earlier attempts (Eqs. 16 & 17) should not come in the results section but should be moved to the introduction.

Answer: We thank the referee for the very comprehensive suggestions. These suggestions contribute to improving the paper. We have turn the Eqs. 16 & 17 into Eqs. 1 & 2 in the Introduction in the revised manuscript.

Thanks and best regards

Attachment:

A_new_derivation_of_the_relationship_k4BJuhY.pdf

---

## Round 2 · Referee Report · Anonymous (Referee 2) · 2021-3-15

Report

I have gone through the revised version of the manuscript and the authors' response to my earlier comments.

I think the manuscript have been considerably revised that it takes into account the earlier concerns raised. Therefore, I recommend publication.

---

## Round 2 · Referee Report · Anonymous (Referee 1) · 2021-3-15

Report

Yes, the requirements of this journal are met

---

## Round 2 · Referee Report · Anonymous (Referee 1) · 2021-3-15

Report

I read the reply of the authors to my original report. I find it and the revised version of the paper as well, satisfactory. For this reason I recommend the revised version for publication

---

## Round 2 · Author Response

Thanks for your kind considerations and referee’s detailed comments and suggestions to improve this work.

We thank the referee's approvement. All the revised parts are written in bold.

~~~~\textbf{Point by point: Reply to the referee comment}\ \ \begin{itshape} The authors report a derivation of a relation between entropy and diffusion coefficient of Brownian point-like particles using Canonical ensemble.

This is a well studied problem, as the authors have said. Similar scaling relations are mentioned in Eqs. 16 \& 17. But they have proposed a new way in deriving such scaling, which could be interesting.

Before recommending, I expect the authors to address the following points: \end{itshape}

\textbf{Answer:} We thank the referee's approvement. In the revised paper, we have performed more details to address the three points based on the referee's suggestions.

\begin{itshape} 1. What is, if any, the advantage of this method over the other methods? Is there any experimental comparison where this method seem to work better? \end{itshape}

\textbf{Answer:} The derivation of Eq.(13) based on the Kolmogorov-Sinai entropy would be showed in APPENDIX A. in APPENDIX B, the formula of the thermodynamic entropy of Brownian particle is derived, but it is hard to analytically solve. Fortunately, Dzugutov et al. have point that Kolmogorov-Sinai entropy, when expressed in terms of the atomic collision frequency, is uniquely related to the thermodynamic excess entropy by a universal linear scaling law\footnote{Dzugutov M., Aurell E., and Vulpiani A. \ 1998, Phys. Rev. Lett. 81, 1762.}. The linear law is not influence the exponential relationship between the diffusion coefficient and the entropy. Our method can give the analytic formula of Kolmogorov-Sinai entropy and make it possible to calculate some more complex model. The Kolmogorov-Sinai entropy is regarded as a measure, for the loss of information about the state of the system, per unit of time. This quantity is more mathematical than physical, so there are not any experimental comparison.

\begin{itshape} 2. The only comparison made so far is in the case of hard sphere model (Eq. 18), where it is argued that for massive Brownian particles (compared to that of the bath molecules), the expressions of entropies are comparable. Is there a quantitative measure of it possible (say, with typical parameter values)?

\end{itshape} One can assume that a system labelled as System $1$ with the volume $V$, only includes $N$ bath particles,which Hamiltonian reads, \begin{equation} \begin{aligned} H= \frac{\textbf{p}^{N}\cdot\textbf{p}^{N}}{2m}+U(\textbf{r}^{N}). \end{aligned} \end{equation} The partition function of this system under canonical ensemble is \begin{equation} \begin{aligned} Z_{1}=\frac{1}{N!h^{dN}}\int e^{-\beta H}d\textbf{p}{1}... d\textbf{p}}d\textbf{r{1}...d\textbf{r}.\ \end{aligned} \end{equation} When one introduces a heavier Brownian particle to join in the system, it is labelled as System $2$, which partition function is \begin{equation} \begin{aligned} Z_{2}=\frac{1}{N!h^{dN}h^{d}}\int e^{-\beta H_{s}}d\textbf{p}{1}... d\textbf{p}}d\textbf{r{1}...d\textbf{r} \end{aligned} \end{equation} one can define the entropy of Brownian particle which equals the difference of entropy of System 2 and System 1. One can obtain }d\textbf{p}d\textbf{x$\Delta \ln Z\equiv \ln Z_{2}-\ln Z_{1}$, based on the formula of the thermodynamic entropy, the thermodynamic entropy of Brownian particle $S_{T}$ reads \begin{equation} \begin{aligned} S_{T}=\frac{kd}{2}[\ln(\frac{2\pi M}{h^{2}\beta})+1]- k\ln[\frac{<e^{\beta\Phi}>}{V}]+k\beta \frac{\partial}{\partial\beta }\ln(<e^{\beta\Phi}>). \label{eq:ss} \end{aligned} \end{equation} Because \begin{equation} \begin{aligned} <e^{A}>&=<1+A+\frac{1}{2}A^{2}+\frac{1}{6}A^{3}+...> \ &=e^{+\frac{1}{2}(<A^{2}>-^{2})+O(A^{3})}. \end{aligned} \end{equation}

\begin{equation} \begin{aligned} &\frac{\partial}{\partial\beta }<\Phi>=<\Phi><H_{0}>-<\Phi H_{0}>\ &=<\Phi>+<\Phi><\Phi>-<\Phi>^{2}-<\Phi U>\ &=<\Phi><\Phi>-<\Phi>^{2}\ \end{aligned} \end{equation}

\begin{equation} \begin{aligned} &<\phi(\textbf{x}-\textbf{r}{i}) U(\textbf{r}}-\textbf{r{j})> \ &=<\phi>(first ~integral ~with~\textbf{r}).\ \end{aligned} \end{equation} So, there is a quantitative measure of it possible but still hard if one can know the values of $<\Phi>$ and $$.

\begin{itshape} 3. Finally, the organization of the paper is somewhat confusing. Earlier attempts (Eqs. 16 \& 17) should not come in the results section but should be moved to the introduction. \end{itshape}

\textbf{Answer:} We thank the referee for the very comprehensive suggestions. These suggestions contribute to improving the paper. We have turn the Eqs. 16 \& 17 into Eqs. 1 \& 2 in the Introduction in the revised manuscript.

\closing{Thanks and best regards}

---

## Round 2 · List of Changes

All the revised parts are written in bold. 1. We have turn the Eqs. 16 \& 17 into Eqs. 1 \& 2 in the Introduction in the revised manuscript.

  1. In section III,we supplement these sentences “{\bf Dzugutov, Aurell and Vulpiani have made the assumption that the Kolmogorov-Sinai entropy can be connected to the conventional thermodynamic entropy\cite{1998PhRvL..81..1762D}. The derivation of Eq.(\ref{eq:Hamiltonian}) based on the Kolmogorov-Sinai entropy would be showed in APPENDIX \ref{sect:Defin}. in APPENDIX \ref{sect:Therm}, the formula of the thermodynamic entropy of Brownian particle is derived, but it is hard to analytically solve. Fortunately, Dzugutov et al. have point that Kolmogorov-Sinai entropy, when expressed in terms of the atomic collision frequency, is uniquely related to the thermodynamic excess entropy by a universal linear scaling law\cite{1998PhRvL..81..1762D}. The linear law is not influence the exponential relationship between the diffusion coefficient and the entropy.}”

  2. we add the sections of APPENDIX A and APPENDIX B.

---

## Editorial Decision

published